# Computational Analysis Identifies Novel Biomarkers for High-Risk Bladder Cancer Patients

**DOI:** 10.3390/ijms23137057

**Published:** 2022-06-24

**Authors:** Radosław Piliszek, Anna A. Brożyna, Witold R. Rudnicki

**Affiliations:** 1Computational Centre, University of Białystok, ul. Konstantego Ciołkowskiego 1M, 15-245 Białystok, Poland; 2Department of Human Biology, Institute of Biology, Faculty of Biological and Veterinary Sciences, Nicolaus Copernicus University, ul. Lwowska 1, 87-100 Toruń, Poland; anna.brozyna@umk.pl; 3Institute of Computer Science, University of Białystok, ul. Konstantego Ciołkowskiego 1M, 15-245 Białystok, Poland

**Keywords:** nonmuscle-invasive bladder cancer (NMIBC), carcinoma in situ (CIS), biomarker identification, optimal feature set selection

## Abstract

In the case of bladder cancer, carcinoma in situ (CIS) is known to have poor diagnosis. However, there are not enough studies that examine the biomarkers relevant to CIS development. Omics experiments generate data with tens of thousands of descriptive variables, e.g., gene expression levels. Often, many of these descriptive variables are identified as somehow relevant, resulting in hundreds or thousands of relevant variables for building models or for further data analysis. We analyze one such dataset describing patients with bladder cancer, mostly non-muscle-invasive (NMIBC), and propose a novel approach to feature selection. This approach returns high-quality features for prediction and yet allows interpretability as well as a certain level of insight into the analyzed data. As a result, we obtain a small set of seven of the most-useful biomarkers for diagnostics. They can also be used to build tests that avoid the costly and time-consuming existing methods. We summarize the current biological knowledge of the chosen biomarkers and contrast it with our findings.

## 1. Introduction

According to the World Cancer Research Fund International, bladder cancer is the 10th most common cancer in the world [1]. It is diagnosed mostly in people over 55 in highly developed countries of southern and western Europe, as well as in North America. Men are more than four times more likely to develop bladder cancer than women. The most commonly mentioned urinary bladder cancer risks, other than being male, are smoking cigarettes, exposure to certain chemicals (such as aromatic amines, polycyclic aromatic hydrocarbons, and chlorinated hydrocarbons and alcohol), having a red meat-rich diet, and being genetically predisposed (reviewed in [1]). Urothelial carcinoma of the bladder is divided into two major groups on the basis of clinical staging with different clinical outcomes and therapy options: non-muscle-invasive bladder cancer (NMIBC) and muscle-invasive bladder cancer (MIBC). MIBCs are aggressive tumors, characterized by a five-year survival rate of less than 50% [2]. Up to 15% of MIBCs are initially diagnosed as NMIBCs that progressed into MIBCs [3]. NMIBC is considered a tumor with a relatively good prognosis since the five-year overall survival rate is about 90% [4]. Unfortunately, NMIBCs are a very heterogeneous tumor group with a high rate of recurrence (up to 70%) and risk of progression to MIBC (up to 20%), despite significant improvement in the adjuvant therapies’ efficacy (reviewed in [5,6,7]). Carcinoma in situ (CIS) belongs to this group and can be diagnosed as a primary or a recurrent tumor. CIS is associated with a poorer prognosis, a higher grade, as well as an elevated risk of recurrence and progression to MIBC [8]. The recurrence rate for CIS is 63–92%, and the progression to MIBC is 50–75%, even when the immunotreatment is applied [9,10]. The current treatment of CIS includes Bacille Calmette–Guérin (BCG) intravesical therapy, but up to 40% of NMIBC patients do not respond to this treatment. In these patients, one of the second-line treatments is cystectomy [11]. However, cystectomy causes side effects, especially in elderly patients. Recent studies identified some predictors of complications, with frailty index score among them [12,13]. Concomitant CIS is also related to a higher recurrence risk and mortality rate [14]. Thus, there is a need to develop accurate methods for the prediction of recurrence and progression in NMIBC, including CIS. Recently, the molecular markers predicting the progression of NMIBC have been identified [15]. However, their testing is based on the evaluation of methylation (GATA2 and TBX3) and mutation status (FGFR3); thus, its usefulness for routine use is rather limited due to the associated cost and labor of the tests [16]. Moreover, there are no specific markers for development of CIS in disease course (CIS-DC). Thus, more exact and accessible models should be developed, and new markers of CIS-DC should be identified.

The goal of the current study is to propose a small, clinically useful set of biomarkers that can be utilized for the stratification of bladder cancer patients into high- and low-risk classes, with respect to the development of CIS in disease course. The study is based on the dataset E-MTAB-4321, first described in [17] and deposited in the ArrayExpress database [18]. The dataset consists mostly of patients with Ta and T1 tumor stages. In the original analysis, the authors applied non-supervised learning to stratify patients into three groups using 119 genetic markers, showing that these three groups differ significantly in the risk of progressing to stage T2+. The original classification was extended in subsequent works by various authors [19,20,21].

The approach proposed in the current study is based on a robust protocol utilizing multiple supervised and non-supervised machine learning methods, including an extensive use of cross validation and resampling.

## 2. Materials and Methods

### Dataset

The E-MTAB-4321 dataset, used in the study, contains clinical and RNA-seq data from 476 patients with early stage urothelial carcinoma, of whom 74 have developed CIS at a certain point of the disease course, whereas 402 were free of CIS during the study period. There are 43,204 genetic markers in this dataset, out of which 4800 have 0 variance, resulting in 38,404 markers actually carrying any information. A summary of the dataset characteristics is present in Table 1 and in Appendix A. For details on data collection, please refer to the original paper by Hedegaard et al. [17].

The analytical protocol is based on supervised feature selection (FS) and supervised classification. In our analysis, we focus on finding markers for predicting the appearance of CIS in disease course.

The following base feature-selection protocol is used. We first identify all informative variables and, therefore, reduce the dimensionality of the problem. Then, we further decrease the dimensionality by clustering similar variables. Finally, we use clusters’ representatives to build machine learning models for the prediction of CIS-DC. Each step is described in detail in the following paragraphs.

In the first step, the variables that carry information about future development of CIS in disease course are identified. To this end, we use the multidimensional feature selection (MDFS) filter, which is based on the information entropy and is available as a library in R [22,23]. The informative variables are identified by computing information entropy conditioned on the knowledge of the descriptive variables and comparing it with the null distribution of information entropy conditioned on the non-informative variables. This metric is called information gain (IG). In this case, we use single-dimensional analysis, which computes maximum IG over multiple (30) random discretizations of continuous variables. The relevance is determined by a *p*-value threshold of 0.05 after applying Holm’s correction [24].

Unlike minimal-optimal approaches to feature selection, all-relevant feature selection does not have a goal of producing the best set of features for model building. On the contrary—the goal is to preserve the information about all relevant variables so that they and their structure can be studied at will. However, this leads to higher complexity for model building and more uncertainty for tooling to discover such structures. To counter this, we used feature clustering to group similar features together.

Similarity is a concept rooted in clustering (and data analysis in general) and is a broad category. For our purpose, we use a correlation coefficient as our similarity metric; precisely, we use the Pearson’s product–moment correlation coefficient ρ. However, for the purpose of applying clustering algorithms, we need a function that can be used as a proper metric—that monotonically describes the similar–dissimilar relation and outputs the penalty associated with dissimilarity. Thus, we apply the following transformation to obtain the function *d*:(1)$$d=1-\rho^{2}$$
which satisfies the properties of a proper metric and describes dissimilarity as a penalty due to lack of correlation. The function *d* is called the dissimilarity function.

We choose hierarchical clustering as our clustering algorithm due to its property of revealing the internal clustering structure. As a method of hierarchical clustering, we evaluated Ward’s minimum variance method as well as the complete linkage method. Of note here is that we applied clustering only to features—not objects nor both objects and features—unlike how clustering and biclustering algorithms are usually used.

We evaluate two ways to choose the representatives to build the classification models. The first is the most commonly applied procedure of working directly with the ranking of features as they are available from the feature-selection method: choosing top-*n* features with the lowest *p*-values. Secondly, we evaluate the effect of hierarchical clustering to *N* clusters and then, analogously, use the ranking to choose one representative from each cluster, basically the top-1 representative from each group.

To evaluate the marker set, we used the Random Forest [25] (RF) implementation available in R’s randomForest package [26] as our target classifier. No tweaks to the default parameters were applied. We used the area under the ROC (receiver operating characteristic) curve, also known as AUROC or even AUC (area under curve), to describe the performance of each built classifier.

While evaluating the stability and generality of the above base protocol, we developed an extended procedure that we present here. We propose the use of cross validation as part of the feature-selection protocol. The entire above-mentioned procedure was run in a stratified 5-fold cross validation with 30 repeats, the direct results of which are presented in Figure 1. Essentially, we have obtained a new ranking from cross validation that allows us to apply the top-*n* procedure while using the count of repetitions as the quality metric. For an overview, see Figure 2.

Furthermore, to estimate the mean and error of our evaluation metric (AUC), we have applied (independently) both external resampling and external cross validation. The resampling procedure consisted of 100 repeats of random sampling with replacement. The omitted objects, called out-of-bag (OOB) objects, were used for verification of the performance of the built models, i.e., for the calculation of the AUC. The cross-validation procedure, on the other hand, was conducted using a stratified 10-fold approach with 30 repeats (independent of the CV inside the procedure). The internal procedure was adjusted to use 10-fold CV as well, to gather enough objects for the MDFS statistic to work well.

Assuming validation with resampling, the full analytical protocol is, thus, as follows (with an overview in Figure 3):repeat 30 times: split data randomly in 5 equal bins (i.e., run 30 repetitions of 5-fold CV) and for each (i-th) bin:
(a)set aside the i-th bin as the test set and create a training set from the 4 remaining bins;(b)identify informative variables in the training set;(c)cluster informative variables using the hierarchical approach and select representatives of each cluster on each clustering level between 2 and 15, utilizing the usual procedure of choosing the most informative one, and:
build an RF model of "CIS in disease course" using those representatives;test the quality of the built model on the test dataset;find cluster representatives at each level that appear most often in the above 150 iterations (30 times 5 iterations), at each level of clustering between 2 and 15;use those representatives for building the final model on the entire dataset:
(a)estimate the confidence intervals of the final models at each number of representatives (between 2 and 15) using the bootstrap approach—repeat 100 times:draw with replacement N patients from the original data, build RF models using the 2 to 15 representative variables;measure the performance of each model using OOB objects;(b)compute the aggregate performance of each model;(c)use the results of the above procedure to propose the relevant markers.

Apart from the above selection of methods, we have verified the final marker set using naive Bayes [27] and logistic regression classifiers, estimating the achievable diagnostic metrics with such simpler classifiers. The details of the naive Bayes classifier are presented in the Appendix A and Appendix B, as it is used as an example simple classifier that is useful for diagnostic personnel.

## 3. Results

The final set of markers was selected after inspecting both the lists of representatives and plots of the AUC in predictive models as a function of the number of markers used. The quality of the predictive models improves with the increasing number of markers used in the model, until it saturates with about seven–nine markers; see Figure 4.

An additional argument for selecting seven markers is the relative stability of the positions of the first seven markers in the list of markers consistently selected in the cross validation; see Figure 5. The first 7 markers appear in the set of the top-7 most-often selected cluster representatives in 150 repeats of the feature-selection procedure, whereas positions of other markers do not rise to the top-7.

The chosen markers maximize the AUC in resampling (see Figure 6) and do not exhibit strong correlations among themselves (as expected from the protocol); see Figure 7. For completeness, we also present the internal quality metric of the protocol in Figure 1 and the details of the selected seven markers in Table 2.

The markers exhibit different directionalities of expression levels between high- and low-risk classes—ADAM28 and TMEM32 expression levels are higher in the low-risk class, while for the other markers, we observe the reverse; see Figure 8.

The diagnostic properties of the models built with the chosen markers are presented in Figure 4. Properties of models from the external cross validation are presented in Table 3. The details of the naive Bayes classifier built on the entire set are reported in Appendix A.

## 4. Discussion

Bladder cancer is one of the most-common cancers in the world [1]; thus, there is a need to develop sensitive methods for the early diagnosis of non-advanced lesions or poor prognosis predictors. Currently, there is a limited number of commercially available tests for bladder cancer diagnosis. The NMP22BC test allows for the diagnosis of non-muscle-invasive bladder cancer and low-grade bladder cancer in urine samples [28]. Recently published data shows that HPLC (high-performance liquid chromatography) of urine could distinguish bladder cancer patients from non-malignant hematuria patients based on chromatographic absorptions and fluorescence peaks [29]. Similarly, fluorescence urine analysis using concentration matrices of synchronous spectra could be useful in bladder cancer diagnosis, allowing to distinguish between cancer patients and heumaturia patients [30]. The new diagnostic strategy could include a label-free optical sensing platform based on DNA strand displacement. Currently, there are no data on bladder cancer detection using this method [31]. Metabolomic analysis is a very promising and useful tool for the identification of biomarkers; it allows for analyses of urine, blood, and tissue samples. The results enable distinguishing between MIBC and NMIBC patients [32]. The aforementioned techniques are aimed at the sensitive and early detection of urinary bladder cancer or at discriminations between MIBC and NMIBC. However, markers allowing for the identification of the risk of CIS development have still not been identified.

CIS of the urinary bladder represents the tumors with high risk of progression to MIBC and metastatic disease [8]. Some data indicate that primary CIS is diagnosed in about 1–3% of newly diagnosed bladder cancers, but some papers report about 20% primary CIS case diagnoses [33,34]. Secondary CIS (detected during follow-up) are diagnosed in about 20% of NMIBC cases [33,35]. Our method allowed for the identification of seven markers related to an increased risk of CIS-DC of urinary bladder cancers. Some of these markers are well-known molecules involved in cancer biology, but some of them are quite unique, with very limited information on their involvement in cancer development and their relationship with tumors.

We identified two markers that are characterized by limited information: DPY19L3-DT (DPY19L3 Divergent Transcript, ENSG00000267213) and E9PMD0 (ENSG00000258472). DPY19L3-DT belongs to the lncRNA class, but there is no information on the function of this molecule in normal and pathological cells and tissues, while the function of E9PMD0 is linked to the cell division and regulation of the attachment of spindle microtubules to kinetochore [36].

We also identified five other markers: ADAM28 (ENSG00000042980), Ras-related C3 botulinum toxin substrate 3 (Rac family small GTPase 3, RAC3, ENSG00000169750), targeting protein for Xenopus kinesin-like protein 2 (TPX2, ENSG00000088325), Ankrd13 family of ubiquitin-interacting motif (UIM)-containing proteins (Ankyrin repeat domain-containing protein 13B, ANKRD13B, ENSG00000198720), and TMEM232. Some of them were previously identified as potential cancer markers or targets for molecular anti-cancer therapies, bladder cancers among them.

ADAM28 belongs to the disintegrin and metalloprotease domain (ADAM) family. Its role in cancers is ambivalent: it promotes cancer cells’ proliferation, survival, migration, and metastasis by affecting neoangiogenesis, epithelial-to-mesenchymal transition, and extracellular matrix degradation, but in the tumor microenvironment it shows strong protective effects against deleterious metastasis dissemination [37]. In bladder cancers, ADAM28 may represent a possible biomarker, since it is overexpressed in bladder transitional cell carcinoma patients and detected in urine [38,39]. In our model, its higher expression was found in patients with low-risk cancers.

Another marker identified by our protocol, RAC3, is involved in neuronal development and in tumor progression, by modulating the organization of the cytoskeleton, cell migration, cell proliferation, and reactive oxygen species production. Its expression was found in different cancers, and it is considered as a marker of poor prognosis, metastasis, and a target for molecular-targeted therapies in some human cancers, such as breast or lung (reviewed in [40]. In our model, the increased expression of RAC3 in high-risk cancers is in line with the existing knowledge and data published by Chen et al. [41]. It indicates that, in bladder cancer, this molecule can be a potential prognostic marker and a target for molecular medicine.

TPX2 is a microtubule-associated protein, involved in the assembly of mitotic spindles and in cell cycles, cell proliferation, and apoptosis [42,43]. TPX2 was found in in silico studies to be related to the risk of the distant metastasis of breast cancers [44]. In bladder cancer, TPX2 is involved in TPX2-mediated phosphorylation of the AURKA-PI3K-AKT axis [45]. In addition, heterogeneous nuclear ribonucleoprotein F, by regulating the TPX2 protein, promotes the cell cycle and proliferation of bladder cancer cells [46]. The proliferation of bladder cancer cells can also be regulated by the interplay between TPX2, p53, and GLIPR1 [47]. In our model, similar to Yan et al. [48], a higher expression of TPX2 was found in high-risk cancers. Thus, we conclude that TPX2 plays an important role in the progression of bladder cancers, including CIS in disease course, and represents a good potential marker for targeted therapy.

ANKRD13B is ubiquitin-binding protein that specifically recognizes and binds Lys-63-linked ubiquitin and that is responsible for the internalization of ligand-activated EGFR [49]. In addition, it is involved in DNA methylation since ANKRD13B (and ANKRD13A and ANKRD13D) form a complex with RNF11 (RING finger protein 11), belonging to the Really Interesting New Gene E3 ligase family (RING) [49,50]. Based on our data, we suggest that ANKRD13B could act as a marker of high-risk bladder cancer, since its expression was significantly elevated in these cancers. It could also be a potential molecular target for anticancer therapies.

TMEM232 is a member of the transmembrane protein family (TMEMs), consisting of more than 300 proteins, being components of cellular membranes [51]. Proteins of this family have differential expression in cancers, but there is limited information on TMEM232. Published data have linked this protein with atopic dermatitis [52,53] or with multiple sclerosis [54]. In our model, the TMEM232 expression pattern was similar to ADAM28, with higher expression in low-risk cancers.

Using externally cross-validated results for the Random Forest classifier and a 75% threshold in our model (Table 3), the fraction of all patients assigned to a high-risk group was 23.6%, and the fraction of all CIS-DC cases assigned to the high-risk group was 49.9%, while the fraction of CIS-DC in a low-risk group was 10.2%, and that in a high-risk group was 32.9%. The fraction of these patients for the 75% threshold, using cross validation and naive Bayes, logistic regression, and Random Forest classifiers are similar, with very promising diagnostic results for Random Forest. The described method could aid clinicians in identifying high-risk bladder cancer (the risk of CIS in disease course). Thus, it offers a diagnostic tool that allows for the personalization of bladder cancer surveillance, more precise treatment option determinations, and the improvement of bladder cancer prognoses.

To summarize, the identified genes can be used as markers of progression in urinary bladder cancers. Moreover, the increased expression of some identified proteins (RAC3, TPX2, ANKRD13B, and TMEM232) indicates their usefulness as potential targets in molecular-tailored therapies. Some of them require more detailed studies since their biological role, especially in cancer, is unknown, or the data are contradictory (ADAM28, TMEM232, DPY19L3-DT, and E9PMD0). We also conclude that, since we identified seven important genes, their evaluation in routine diagnostic procedures is possible using immunohistochemistry or in situ hybridization. Such a panel would not burden laboratories with high costs and labor. Finally, a ready classifier based on naive Bayes technique is presented in the Appendix A and Appendix B, along with an example calculation to enable the research and diagnostics communities to readily analyze applicable data.

## Figures and Tables

**Figure 1 ijms-23-07057-f001:**
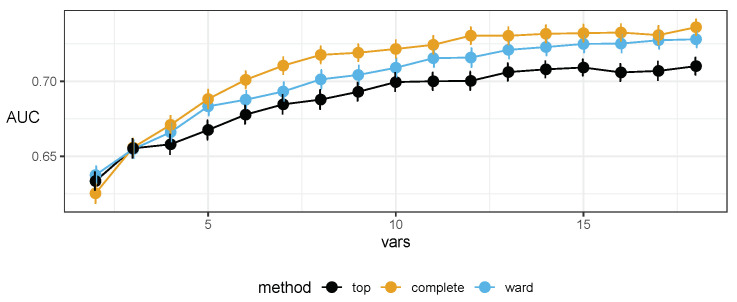
Plots of the area under the receiver operating characteristic curve (AUC) of the Random Forest classifiers, using markers selected by the top-*n* approach and two variants of hierarchical clustering inside our proposed protocol (complete linkage and Ward’s criterion). These results were obtained without external cross validation (CV) or resampling, but from inside of the protocol itself (that is, under the protocol’s internal CV). Error bars denote the standard error.

**Figure 2 ijms-23-07057-f002:**
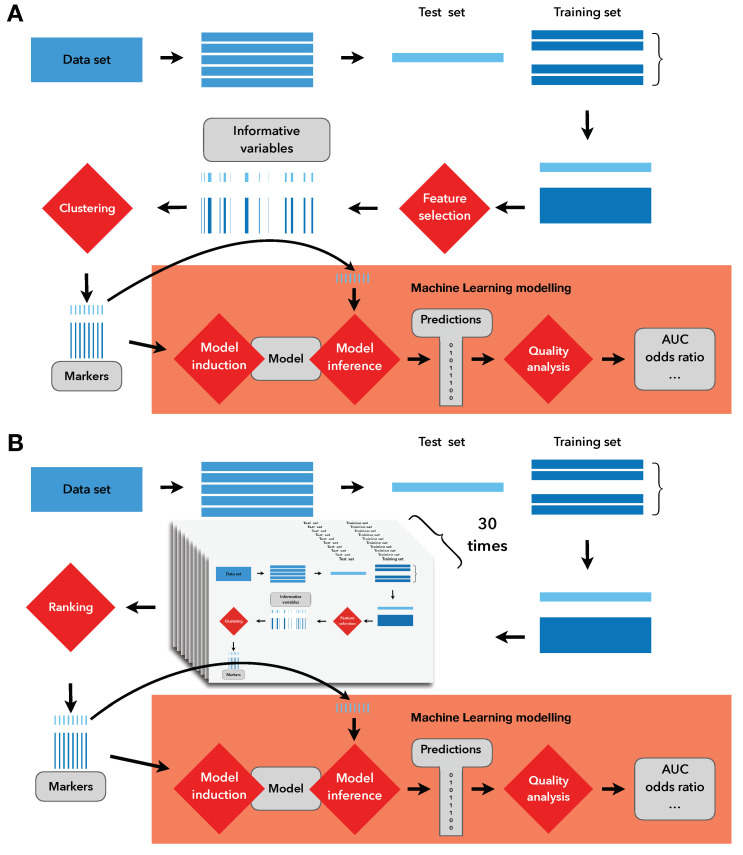
Depiction of a single run of the base protocol (**A**) and the proposed protocol (**B**). In (**A**), the clustering is used directly to obtain markers and build models on them. In (**B**), the (**A**) part is replicated, except for the highlighted part regarding model building and evaluation. Instead, the results of (**A**) are used to build a ranking of the most-commonly chosen variables, which then are used for model building and evaluation.

**Figure 3 ijms-23-07057-f003:**
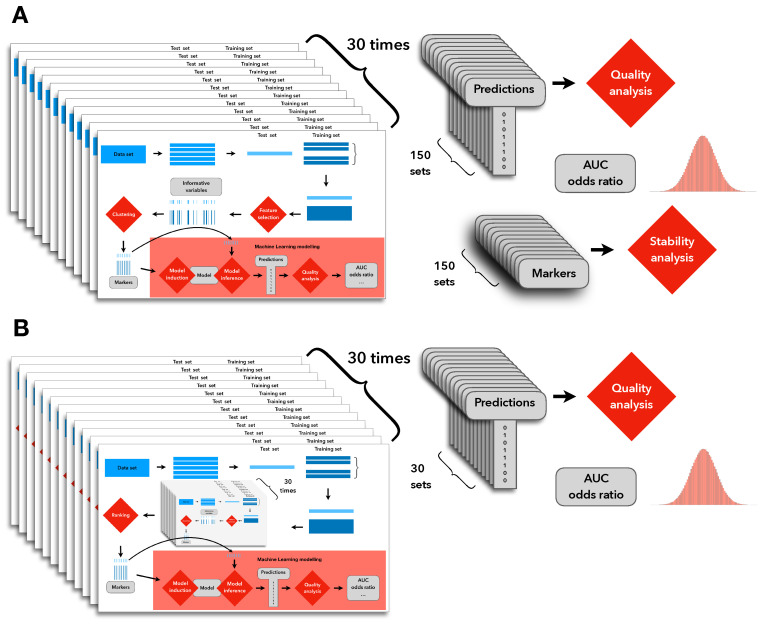
Depiction of the final evaluation. Both variants (**A**,**B**) from Figure 2 were evaluated, respectively. An external cross validation was used to obtain the mean and standard deviation of quality metrics (AUC, odds ratio). The evaluation of the stability of variant (**A**) (the base protocol) prompted us to create and apply variant (**B**) (the proposed protocol).

**Figure 4 ijms-23-07057-f004:**
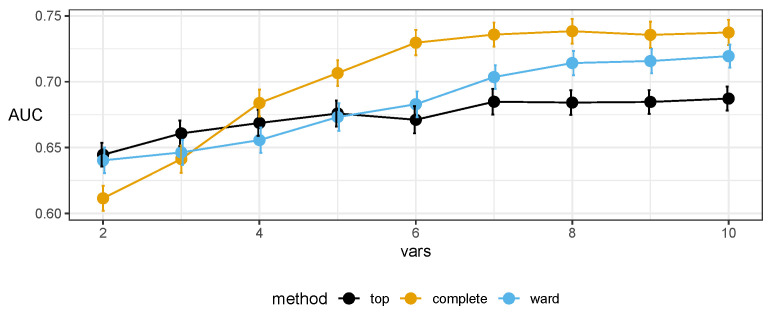
Plots of area under the receiver operating characteristic curve (AUC) of Random Forest classifiers using markers selected by the top-*n* approach and two variants of hierarchical clustering inside our proposed protocol (complete linkage and Ward’s criterion). These results were obtained in external 10-fold cross validation (CV). Internally, for the protocol, 10-fold CV was used to ensure enough samples. Error bars denote the standard error. The complete linkage variant exhibits the desired behavior, achieving the best results earliest, with a plateau starting at 7.

**Figure 5 ijms-23-07057-f005:**
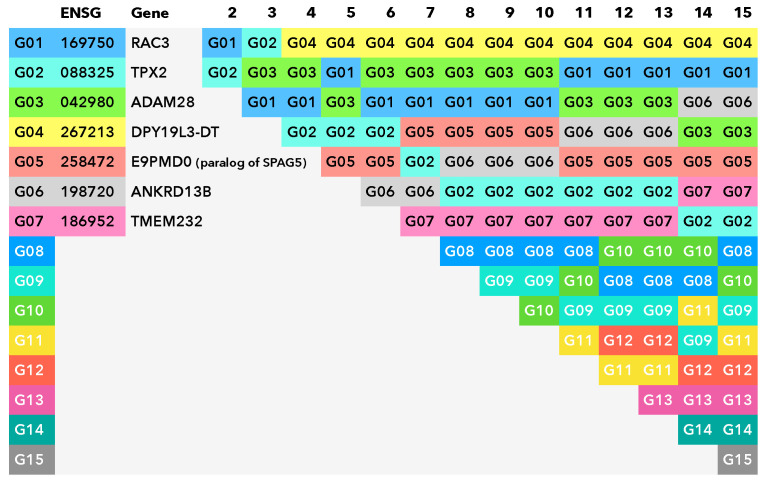
Most-representative markers at different clustering levels in 150 repeats of hierarchical clustering procedure. The first 3 columns show the order in which markers are included in the representative set, when the number of representatives is increased by 1—from 2 to 15. The Ensemble code of each marker, with 5 leading zeros removed, is shown in column 2, and the gene name corresponding to the marker is shown in column 3. In the remaining columns, the markers that are most often selected as representatives in 150 repeats are shown, and their positions within the column corresponds to the frequency of selection of a given marker as the representative (higher position—higher frequency).

**Figure 6 ijms-23-07057-f006:**
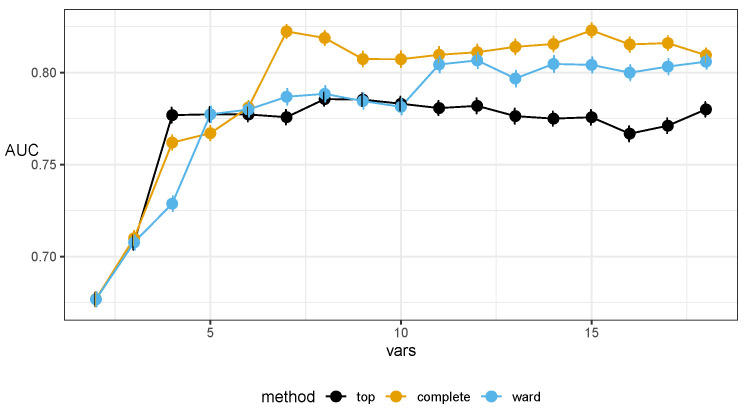
Plots of the area under the receiver operating characteristic curve (AUC) of Random Forest classifiers, using markers selected by the top-*n* approach and two variants of hierarchical clustering inside our proposed protocol (complete linkage and Ward’s criterion). These results were obtained in 100 runs of resampling of the standard protocol, as described in the paper body. Error bars denote the standard error. The complete linkage variant again exhibits the desired behavior, achieving the best results earliest, with the plateau starting at 7.

**Figure 7 ijms-23-07057-f007:**
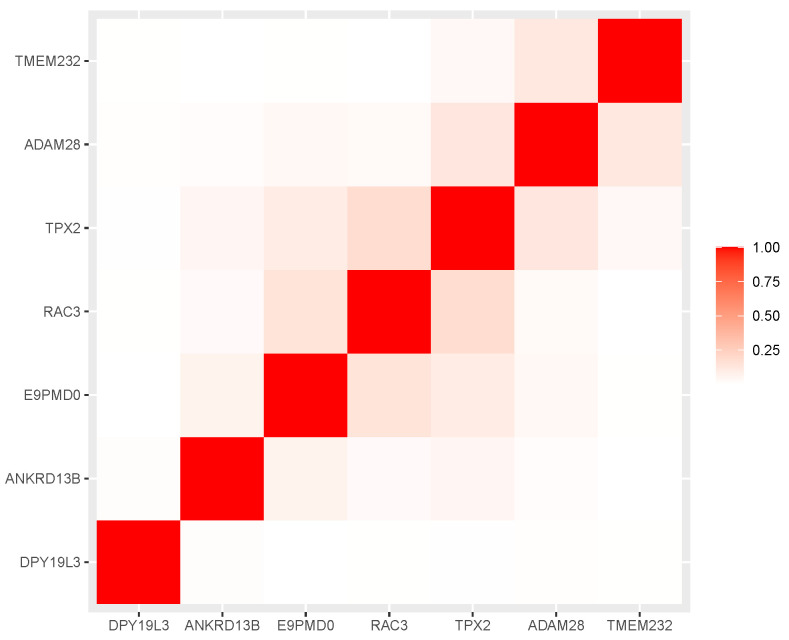
Heatmap of the correlation square of the chosen genes’ expression levels. The darker (more saturated) the square, the higher the level of correlation.

**Figure 8 ijms-23-07057-f008:**
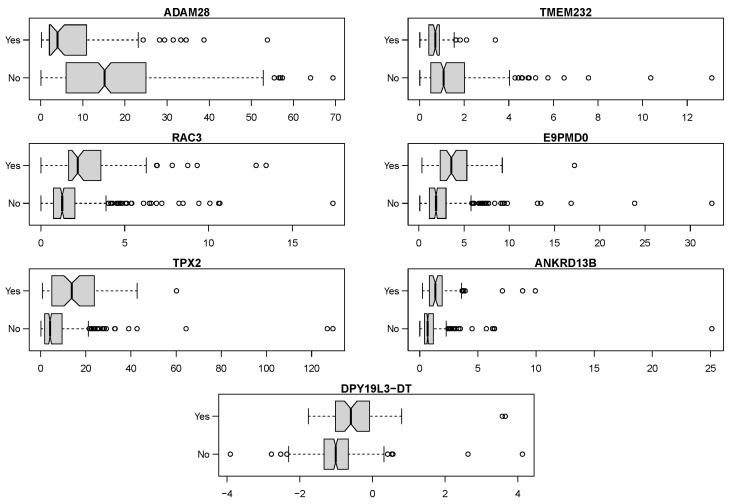
Boxplots of expression levels of selected markers comparing samples with CIS in disease course and those without it. Values of “DPY19L3-DT” are presented after applying a logarithm operation to be able to show the difference. Others are plotted verbatim. It can be seen that low expression levels of ADAM28 and TMEM232 increase the risk of CIS in disease course, while the 5 other variables exhibit the inverse behavior.

**Table 1 ijms-23-07057-t001:** Dataset characteristics. BCG—Bacillus Calmette–Guérin vaccine. PUNLMP—papillary urothelial neoplasm of low-malignant potential. CIS—carcinoma in situ (in the table as a stage of tumor when its sample was taken). More details on the dataset are available in the Appendix A and Appendix B.

Female	109	W/o cystectomy	444	W/o BCG treatment	388
Male	367	W/cystectomy	32	W/BCG treatment	88
CIS	3	High grade	192	W/o CIS in disease course	402
Ta	345	Low grade	277	W/CIS in disease course	74
T1	112	PUNLMP	7		
T2-4	16				

**Table 2 ijms-23-07057-t002:** Details of the selected genes. The repetitions shown are for the case of selecting 7 clusters. There were 150 (30 times 5) trials, and thus, 150 is the upper bound for repetitions. IG stands for information gain, and here, it is the maximum IG computed by the MDFS library in 1D on the entire set (30 random discretizations were used). Label is a shortened version of gene name, used for identification purposes in other parts of the paper.

	Ensembl Gene ID	Repetitions	IG	Gene Name	Label
1	ENSG00000267213	114	29.3	DPY19L3-DT	DP
2	ENSG00000042980	66	35.5	ADAM28	AD
3	ENSG00000169750	64	34.0	RAC3	RA
4	ENSG00000258472	52	24.9	E9PMD0 (paralog of SPAG5)	E9
5	ENSG00000088325	50	29.0	TPX2	TP
6	ENSG00000198720	47	28.1	ANKRD13B	AN
7	ENSG00000186952	38	30.0	TMEM232	TM

**Table 3 ijms-23-07057-t003:** Externally cross-validated results for the Random Forest classifiers. The first column defines the threshold set in the classifiers, separating low- and high-risk groups. The second column displays the fraction of all patients assigned to a high-risk class (HRC) at a given threshold. Analogously, the third column presents the fraction of all CIS-DC cases assigned to the high-risk class. Two next columns present the fraction of CIS-DC in a low- and high-risk class (LRC and HRC), respectively. Similarly, columns six and seven present the odds of CIS-DC in an LRC and HRC, respectively. Finally, the DOR column displays the diagnostic odds ratio between the HRC and LRC, and RR displays the risk ratio between these classes. The comment on the cutoff and share of patients in the HRC from Table 4 applies here as well.

Cutoff	Share of	Share of	Risk	Risk	Odds	Odds	DOR	RR
	Patients	CIS-DC	for	for	for	for		
	in HRC	in HRC	LRC	HRC	LRC	HRC		
50%	52.0%	78.8%	6.9%	23.6%	0.07	0.31	4.2	3.4
75%	23.6%	49.9%	10.2%	32.9%	0.11	0.49	4.3	3.2
90%	8.3%	24.6%	12.8%	46.2%	0.15	0.86	5.9	3.6

**Table 4 ijms-23-07057-t004:** Cross-validated results for the three classifiers, built using the seven selected markers. Stratified 5-fold cross validation, repeated 30 times, was used. Column headings are as in Table 3. Please note that the observed inconsistency between the cutoff and the share of patients in the HRC stems from the application of cross validation—the share is averaged over all folds from all iterations, while the cutoff is established using the entire dataset a priori.

Cutoff	Share of	Share of	Risk	Risk	Odds	Odds	DOR	RR
	Patients	CIS-DC	for	for	for	for		
	in HRC	in HRC	LRC	HRC	LRC	HRC		
Naive	Bayes							
50%	49.6%	79.5%	6.3%	24.9%	0.07	0.33	4.9	3.9
75%	25.5%	57.7%	8.8%	35.2%	0.10	0.54	5.6	4.0
90%	11.0%	33.9%	11.5%	47.9%	0.13	0.92	7.0	4.1
Logistic	Regression							
50%	49.9%	86.0%	4.4%	26.8%	0.05	0.37	8.0	6.2
75%	25.1%	57.4%	8.8%	35.5%	0.10	0.55	5.7	4.0
90%	10.6%	33.6%	11.5%	49.1%	0.13	0.97	7.4	4.3
Random	Forest							
50%	49.1%	86.6%	4.1%	27.4%	0.04	0.38	8.9	6.7
75%	26.0%	68.0%	6.7%	40.6%	0.07	0.68	9.5	6.0
90%	10.2%	35.5%	11.2%	54.3%	0.13	1.19	9.5	4.9

## Data Availability

In the paper, we used a publicly available dataset E-MTAB-4321 available at https://www.ebi.ac.uk/arrayexpress/experiments/E-MTAB-4321/.

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
