# Peer review of "Computational Analysis Identifies Novel Biomarkers for High-Risk Bladder Cancer Patients"

_ijms, 2022, doi:10.3390/ijms23137057_

Round 1

Reviewer 1 Report

The mathematical and statistical calculations given in the manuscript could be interesting and useful. However, the manuscript needs serious improvements:

Material and Methods

Information on the processed set of individuals is given in only one sentence (l. 59 - l. 62) and Table 1. The presented data in Table 1 raise questions: How do the data on Female 109, Male 367 correlate with other data in the table (in the relevant row / columns)? Such a question also arises for information about cystectomy, treatment, CIS during the disease. What does CIS 3 mean?

Although the authors broadly describe the mathematical - statistical methods used, they are often general. Figure 1 is school type and brings nothing new. Figure 2 should be more specified or replaced by short text.

Results

The results are difficult to evaluate due to confusing information in M&M. Please modify the table and figure captions to bring as much information as possible.

Discussion

Please specify the interpretation of your individual results and their contribution to science and practice, in comparison with already published works, which should be adequately added to the citations, eg:

di Meo, NA et al.: 2022 | INTERNATIONAL JOURNAL OF MOLECULAR SCIENCES 23 (8); Dzubinska, D.  et al.:  2021 | DIAGNOSTICS 11 (10); Zhang, YY; et al.: 2019 | COMPUTATIONAL BIOLOGY AND CHEMISTRY 78 , pp.448-454; Kollarik, B et al.: 2018 | NEOPLASMA 65 (2) , pp.234-241; Choi, H.S.; et al. 2010 | Korean J. Urol. 2010, 51, 88–93.

Author Response

We thank the reviewer for their thorough review. We have approached each comment with due diligence and made appropriate changes in the manuscript. In the following reply, we present our responses and the relevant changes per each comment.

Regarding the comment about the Materials and Methods section, we have included additional tables summarising the data distribution in the new appendix and explained that CIS as a stage means that CIS was present already when the samples were collected. Furthermore, we have made clearer the thought process associated with the protocol invention, resulting in (among other minor changes) the following new paragraph separating the base from the final protocol:

While evaluating the stability and generality of the above base protocol, we established another procedure that we present here. What we propose is the use of cross-validation as part of the feature selection protocol. The entire above procedure was run in a stratified 5-fold cross-validation with 30 repeats, direct results of which are presented in Figure 7. Essentially, we have obtained a new ranking from cross-validation that allows us to apply the top-n procedure but using the count of repetitions as the quality metric.

We have also replaced the figures with new, more detailed ones. We believe, based on our experience, that such figures improve the understanding of the main text which is otherwise self-sufficient already.

Regarding the comment about the Results section, we believe the improvements made as part of the reply to the previous comment make the results easier to evaluate. To this end, we have also expanded the captions in the Results section to include more details at hand. For example, the first figure in the Results section has the following caption now:

Plots of area under the receiver operating characteristic curve (AUC) of Random Forest classifiers using markers selected by top-$n$ approach and two variants of hierarchical clustering inside our proposed protocol (complete linkage and Ward's criterion). These results were obtained in external 10-fold cross-validation (CV). Internally, for the protocol, 10-fold CV was used to ensure enough samples. Error bars denote the standard error. The complete linkage variant exhibits the desired behaviour, achieving the best results earliest, with plateau starting at 7.

Regarding the comment about the Discussion section, we have added the following opening information in that section:

Bladder cancer is one of the most common cancers in the world [1], thus there is a need to develop sensitive methods for early diagnosis of non-advanced lesions or poor prognosis predictors. Currently, there is a limited number of commercially available tests for bladder cancer diagnosis. The NMP22BC test allows for diagnosis of non-muscle-invasive bladder cancer and low-grade bladder cancer in urine samples [28]. Very recent published data showed that HPLC (high-performance liquid chromatography) of urine could distinguish bladder cancer patients from non-malignant hematuria patients based on chromatographic absorptions and fluorescence peaks [29]. Similarly, fluorescence urine analysis using concentration matrices of synchronous spectra could be useful in bladder cancer diagnosis to distinguish cancer patients with hematuria patients [30]. The new diagnostic strategy could include a label-free optical sensing platform based on DNA strand displacement. Currently, there is no data on bladder cancer detection using this method [31]. The very promising and useful tool for identification of the new biomarkers is metabolomic analysis, allowing for analysis of urine, blood, tissue samples. This technique allows for distinguishing MIBC and NMIBC patients [32]. The aforementioned techniques are aimed for sensitive and early detection of urinary bladder cancer or discrimination MIBC and NMIBC. However, there is still lack of markers allowing for identification of risk of CIS development.

References:

  1. Choi, H.S.; Lee, S.I.; Kim, D.J.; Jeong, T.Y. Usefulness of the NMP22BladderChek test for screening and follow-up of bladder cancer. Korean Journal of Urology 2010, 51, 88–93.
  2. Džubinská, D.; Zvarík, M.; Kollárik, B.; Šikurová, L. Multiple Chromatographic Analysis of Urine in the Detection of Bladder Cancer. Diagnostics 2021, 11, 1793.
  3. Kollarik, B.; Zvarik, M.; Bujdak, P.; Weibl, P.; Rybar, L.; Sikurova, L.; Hunakova, L. Urinary fluorescence analysis in diagnosis of bladder cancer. Neoplasma 2018, 65, 234–41.
  4. Zhang, Y.;Wang, L.;Wang, Y.; Dong, Y. Label-free optical biosensor for target detection based on simulation-assisted catalyzed hairpin assembly. Computational Biology and Chemistry 2019, 78, 448–454.
  5. di Meo, N.A.; Loizzo, D.; Pandolfo, S.D.; Autorino, R.; Ferro, M.; Porta, C.; Stella, A.; Bizzoca, C.; Vincenti, L.; Crocetto, F.; others. Metabolomic Approaches for Detection and Identification of Biomarkers and Altered Pathways in Bladder Cancer. International

Journal of Molecular Sciences 2022, 23, 4173.

Reviewer 2 Report

Interesting study aimed to identify biomarkers in the field of bladder cancer - carcinoma in situ. Overall, the study sounds scientifically well and evaluated an important urological issue. I have appreciated the study.

In have only one suggestion (minor) regarding Introduction.

Please, underline the clinical issue about CIS: in absence of BCG response radical cystectomy is suggested as treatment. Such surgery is not devoid of complications and several studies have been carried out to detect predictors of complications. Thus, please consider reporting the following two studies in introduction with the aim to summarize the above clinical issue: PMID: 31402279 and PMID: 23942944

Author Response

We thank the reviewer for their review and the kind words. In reply to the comment made, we have added the following information in the Introduction section: 

The current treatment of CIS includes Bacille Calmette–Guérin (BCG) intravesical therapy, but up to 40% of NMIBC patients do not respond to this treatment. In these patients, one of second-line treatments is cystectomy [11]. However, cystectomy exhibits side effects, especially in elderly patients. Recent studies identified some predictors of complications, with frailty index score among them [12,13].

References:

  1. Lebacle, C.; Loriot, Y.; Irani, J. BCG-unresponsive high-grade non-muscle invasive bladder cancer: what does the practicing urologist need to know? World Journal of Urology 2021, 39, 4037–4046.
  2. De Nunzio, C.; Cicione, A.; Izquierdo, L.; Lombardo, R.; Tema, G.; Lotrecchiano, G.; Minervini, A.; Simone, G.; Cindolo, L.; D’Orta, C.; others. Multicenter analysis of postoperative complications in octogenarians after radical cystectomy and ureterocutaneostomy: the role of the frailty index. Clinical Genitourinary Cancer 2019, 17, 402–407.
  1. Cantiello, F.; Cicione, A.; Autorino, R.; Salonia, A.; Briganti, A.; Ferro, M.; De Domenico, R.; Perdonà, S.; Damiano, R. Visceral obesity predicts adverse pathological features in urothelial bladder cancer patients undergoing radical cystectomy: a retrospective cohort study. World Journal of Urology 2014, 32, 559–564

Round 2

Reviewer 1 Report

The name of the vertical axis in Figures 3, 5, 7 should be in capital letters AUC, not auc.

Author Response

We thank the reviewer for being observant. We fixed the axis' name.